# Targeting Biofilm of MDR *Providencia stuartii* by Phages Using a Catheter Model

**DOI:** 10.3390/antibiotics10040375

**Published:** 2021-04-02

**Authors:** Chani Rakov, Shira Ben Porat, Sivan Alkalay-Oren, Ortal Yerushalmy, Mohanad Abdalrhman, Niv Gronovich, Lina Huang, David Pride, Shunit Coppenhagen-Glazer, Ran Nir-Paz, Ronen Hazan

**Affiliations:** 1Institute of Dental Sciences, School of Dentistry, Hebrew University of Jerusalem, Jerusalem 91120, Israel; chani2061@gmail.com (C.R.); shira.benporat@mail.huji.ac.il (S.B.P.); sivan.alkalay@mail.huji.ac.il (S.A.-O.); ortal6991@gmail.com (O.Y.); niv7804@gmail.com (N.G.); Shunitc@ekmd.huji.ac.il (S.C.-G.); 2Hadassah-Hebrew University Medical Center, Department of Clinical Microbiology and Infectious Diseases, The Faculty of Medicine, Hebrew University of Jerusalem, Jerusalem 91120, Israel; mohanad.abdalrhman@mail.huji.ac.il (M.A.); NIRPAZ@hadassah.org.il (R.N.-P.); 3Department of Pathology, University of California, San Diego, CA 92093, USA; lsh007@health.ucsd.edu (L.H.); dpride@health.ucsd.edu (D.P.)

**Keywords:** phage therapy, antibiotic resistance, *Providencia stuartii*, *Providencia rettgeri*, catheter infections

## Abstract

*Providencia* spp. are emerging pathogens mainly in nosocomial infections. *Providencia stuartii* in particular is involved in urinary tract infections and contributes significantly to the high incidence of biofilm-formation in catheterized patients. Furthermore, recent reports suggested a role for multiple drug resistant (MDR) *P. stuartii* in hospital-associated outbreaks which leads to excessive complications resulting in challenging treatments. Phage therapy is currently one of the most promising solutions to combat antibiotic-resistant infections. However, the number of available phages targeting *Providencia* spp. is extremely limited, restricting the use of phage therapy in such cases. In the present study, we describe the isolation and characterization of 17 lytic and temperate bacteriophages targeting clinical isolates of *Providencia* spp. as part of the Israeli Phage Bank (IPB). These phages, isolated from sewage samples, were evaluated for host range activity and effectively eradicated 95% of the tested bacterial strains isolated from different geographic locations and displaying a wide range of antibiotic resistance. Their lytic activity is demonstrated on agar plates, planktonic cultures, and biofilm formed in a catheter model. The results suggest that these bacteriophages can potentially be used for treatment of antibiotic-resistant *Providencia* spp. infections in general and of urinary tract infections in particular.

## 1. Introduction

*Providencia stuartii* and *Providencia rettgeri* are members of the Enterobacteriaceae family [1]. The involvement of these opportunistic pathogens in human health-care-associated infections was recognized long ago, and indeed several health care outbreaks of *Providencia* spp. have been reported [2,3]. While it is considered uncommon, recent reports from various countries established the emergence of multiple drug resistant (MDR) isolates of *Providencia* species [4,5,6] and demonstrated their ability to disseminate amongst patients in hospital settings leading to nosocomial outbreaks with treatment challenges and complications [7,8,9].

Various studies showed that *Providencia* species that are involved in urinary tract infections are also biofilm-forming [10,11]. The urinary tract provides a unique niche for the colonization of this bacterium [12]; therefore, it is highly prevalent in bacteriuric long-term-catheterized patients [10,13]. Infections by *P. stuartii* may reach mortality rates of 30% in nosocomial outbreaks [4]. Recent outbreaks of *Providencia rettgeri* have also raised public health concerns [7].

*Providencia* spp. harbor several different virulence factors such as chromosomal *ampC* gene and plasmid-mediated resistance mechanisms such as Extended-spectrum beta-lactamases (ESBLs) and Metallo-beta-lactamases (MBLs) [14]. Additionally, intrinsic inherited resistance to aminopenicillins, colistin, beta-lactams such as early generation of cephalosporins, carbapenems, as well as aminoglycosides and many other antibiotic classes can also be found in this species [6,15,16,17]. Moreover, *P. stuartii* can create floating communities that facilitate adhesion and biofilm production leading to biofilm-associated infections [10]. These virulence factors make *Providencia* spp. difficult to treat [18].

Taking into account the increasing prevalence of *Providencia* spp. cases, their role in hospital-acquired infections, and the emergence of bacterial resistance, it is clear that additional or alternative tools for antibiotic treatments are necessary.

One such promising modality is phage therapy [19]. Indeed, the concept of harnessing bacteriophages to eradicate resistant infections has been re-discovered in Western medicine and is gathering significant attention worldwide as a new tool to combat infectious diseases [20]. In the last few years, reports of compassionate-use cases and early stage clinical trials are published, suggesting that bacteriophages against *P. stuartii* might also serve as a potential treatment. Nevertheless, to the best of our knowledge, the number of available *Providencia* spp. phages is rather limited [21] and their description and analysis is incomplete [22].

Here, we describe the isolation and characterization of 12 lytic and 5 temperate bacteriophages including full genome sequencing performed by the Israeli Phage Bank (IPB) [23]. The phages target multi-drug resistant clinical isolates of *Providencia* spp. across the world. We tested the treatment potential of these phages, by demonstrating their ability to eradicate biofilms built in a catheter model. As the phages present high efficiency in killing bacteria, we suggest that they might be useful as therapeutic agents.

## 2. Results

### 2.1. Phage Isolation and Assessment of Host-Range Coverage

A screening of sewage samples for the presence of potential anti-*Providencia* species bacteriophages was performed as described in “Materials and Methods” (phage isolation and propagation). After a preliminary enrichment step, and subsequent spotting of the lysate on bacterial lawns, numerous samples resulted in reproducible lysis of bacterial cells and exhibited clear plaques. A total of 17 phages were isolated from single plaques (Table 1) and further propagated in liquid culture as detailed in “Materials and Methods”, showing lysis after an overnight incubation in liquid culture. The phages were evaluated for host range activity by the drop test. As the goal here was to supply a satisfying coverage of *Providencia* spp., strains that showed sensitivity to one of the phages were excluded from the subsequent screening cycles. The results demonstrated that despite the bacteria’s wide-range antibiotic resistance, the phages targeted 39 out of 41 clinical isolates, providing a coverage of 95% of the isolates tested here (Figure 1). 

### 2.2. Genome Analysis

Bacteriophages’ genomes were subsequently sequenced and annotated. Phages’ names, their putative lifecycle type, topology, genome size, phylogeny, and GenBank accession number are denoted in Table 1.

Alignment of the genomes revealed a high diversity between the phages reported here. According to the NCBI database, the lytic phages PSTCR4, PSTCR6, PSTCR7 and Kokobel2 do not have high similarity to known phages. PSTCR4 and PSTCR7 do not overlap any other published genomes (0% query cover), those two phages have high similarity to each other with 93% coverage, 97% identity, and about 9% coverage with Kokobel2. PSTCR6 have only 20% coverage with the known Shigella phage vB_SdyM_006 (MK295204.1). The temperate phage PSTCR7lys which has some similarities with genomic regions of known *Providencia* strains was found to be unique and it seems that it belongs to a new phage subfamily. Some of the phages have genomic similarity to phages with different hosts; Kokobel1 genome displays similarity to *Proteus* phages and PSTRCR_121, PSTRCR_127 and PSRCR_128 display similarity to *Morganella* phages. The phylogenetic tree of our phages compared with the few published *Providencia* phages and other relevant phages (obtained from the NCBI database) is provided in Figure 2. 

As potential candidates for phage therapy treatment, the phages were also scanned for virulence factors and antimicrobial resistance genes using Abricate (https://github.com/tseemann/abricate, release 24, accessed on 28 March 2020) with all its databases: NCBI, CARD, ARG-ANNOT, Resfinder, EcOH, MEGARES, PlasmidFinder, Ecoli_VF and VFDB. The phages were found to be free of virulence factors and antibiotic resistance genes (data not shown).

### 2.3. Characterization of Phage Lytic Activity

To further test the phages’ applicability against *Providencia*, lytic activity was assessed on agar plates, planktonic cultures, and biofilm formed on a catheter. The isolate P.st 8 (with its phages PSTCR4 and PSTCR6), was chosen to represent *P. stuartii* isolates for deeper investigation, since this bacterium was found to be one of the most resistant isolates (8 out of 10 antibiotics, see Table 2 in “Materials and Methods”). P.rett10 (with its phage PSTCR2), was selected arbitrarily, to represent *P. rettgeri* isolates. 

Consistent with the fact that these phages were confirmed to be genotypically distinct from each other, the infectivity patterns of the different phages were found to be also unique phenotypically: bacteriophages (PSTCR4, PSTCR6, and PSTCR2) formed plaques of different sizes and transparencies levels, and some of them had a halo region surrounding the plaque (Figure 3a–c).

### 2.4. Electron Microscopy Visualizing

To observe and determine the geometric structure and morphological characteristics of PSTCR4, PSTCR6 and PSTCR2, these phages were visualized under a Transmission Electron Microscope (TEM). TEM microscopy showed that PSTCR4 has a hexagonal capsid with a measured diameter of 105/75 nm and a tail length of 61 nm with long fibers (Figure 3d). PSTCR6 has a hexagonal capsid with a measured diameter of 101/66 nm and a tail length of 60 nm with long-tailed fibers. In most images obtained, phages were attached or were hidden between vesicles secreted by the bacteria. (Figure 3e). PSTCR2 has a hexagonal capsid with a measured diameter of 55/50 nm and a very short tail length of 18 nm (Figure 3f). 

### 2.5. Kinetic of Lytic Activities in Liquid Culture

Phages also exhibited different inhibition patterns, as observed by the different kinetics of bacterial growth curves in the presence of the different phages: at the *P. stuartii* logarithmic stage, while treatment of PSCR4 showed an effect on bacterial growth only after 10 h with the best lysis after 23 h followed by a regrowth of the bacteria, PSTCR6 kept bacteria from growing for the first 10 h, followed by a regrowth that was disrupted by another lysis peak towards the stationary stage (Figure 4a). Cultures treated with both phages (PSTCR4 and PSTCR6) displayed a synergistic effect and kept bacteria from growing throughout 50 h of the experiment (Figure 4a). At the stationary phase of P.st 8, PSTCR4 exhibited a slight lytic effect after 40 h of growth, while PSTCR6 acted earlier, after 27 h, and no additive effect was obtained when both phages were added (Figure 4b). Impressive actual lysis of *P. rettgeri* cultures was observed by PSTCR2 totally inhibiting logarithmic bacterial growth throughout the time of the experiment (Figure 4c) and reducing stationary culture turbidity 2.4-fold (Figure 4d). 

### 2.6. Biofilm Degradation Activity

Biofilm infections pose a huge challenge to human health as they are very difficult to treat by most antibacterial strategies, including conventional antibiotics [24]. Thus, an assessment of phages’ efficacy against *P. stuartii* biofilm formed on a catheter was performed using one of the most resistant isolates, P.st 8 and its two lytic phages (PSTCR4 and PSTCR6) as a representative model. Latex and silicone catheters were cut to ring-shaped pieces and infected with bacteria at the logarithmic stage, once the biofilm was established, phage was inoculated and incubated overnight. The cultures which contained phage were less turbid compared to the cultures without phage as can be seen in Figure 5a. 

The biofilm degradation activity was assessed by biomass staining, viable counting of bacteria, SEM and confocal microscopy as described in detail in “Materials and Methods”. Crystal violet biofilm biomass evaluation showed a 1.9- and 2-fold reduction in 3-day-old *P. stuartii* biofilms built on latex or silicone catheters, respectively, following 24 h of treatment, in comparison with the untreated biofilm controls (Figure 5b). To exclude the interference of bacterial debris and to verify the eradication of live bacteria, we further tested the viable counts of the biofilm itself. Biofilms were disrupted by sonication and viable counts of live bacteria were performed. Viable counts showed a 2.86- and 2.46-log growth reduction by phage exposure in latex and silicone catheters, respectively, compared to untreated biofilms (Figure 5c). 

To further assess phage efficacy against *P. stuartii* biofilm, SEM was used to visualize the effect of phage on a 3-day-old biofilm. Figure 5d shows the surface topography of naked latex and silicone catheters and biofilms formed on those catheters. The surface area of the latex catheter appeared to have protrusions and spherical structures. Exposure to phage revealed an extensive effect of bacterial lysis leaving cellular debris and almost no trace of the biofilm. (Figure 5d). 

Finally, catheters were stained with the fluorescence stain syto and visualized by a confocal laser microscope. A significant difference was observed in biofilm formed on a silicone catheter treated with phage compared with the untreated controls (Figure 5e). This method was not appropriate for the latex catheters as for an unknown reason, the catheter itself absorbs the stain more efficiently than the biofilm treated with phage (data not shown).

## 3. Discussion

This article reports the isolation, characterization, and full genome sequencing of 17 novel phages targeting *P. stuartii* and *P. rettgeri* clinical isolates, providing a satisfying coverage of 95% of the host strains tested. The phages were active against *Providencia* isolates across the world, portending for their activity against yet unknown strains.

Some of the phages’ genomes were found to be unique exhibiting distinct differences compared to each other and compared to other known genomes in the NCBI databases. In our experience, this is rather unusual in comparison to other phage-analyses we previously conducted [23].

The extreme specificity of phages requires a personalized-medicine approach. Therefore, characterization of all 17 phages with all 41 bacterial hosts is tedious and irrelevant. When a candidate patient is ready for treatment, we screen all our phages to assess their killing abilities and characterize the most suitable one on the patient’s specific isolate. Therefore, here, characterization was carried out on representative phage and hosts. In vitro characterization of three of the lytic phages (PSTCR2, PSTCR4 and PSTCR6) showed killing capabilities in solid and liquid cultures in various patterns and levels of efficiencies. 

Consistent with the results for the planktonic cultures, the phages PSTCR4 and PSTCR6 exhibited an efficient reduction of well-established *P. stuartii* biofilms formed in catheter models using various methods. These results, together with the finding that the phages are free of virulence factors and resistance genes, suggest the potential of those candidate phages as reliable therapeutic agents for phage therapy against *P. stuartii* biofilms in bacteriuric catheterized patients. 

Nevertheless, as the ability to disrupt biofilms is not obvious for any phage, we postulate that the phages’ ability to remove biofilm should also be tested by adding small volumes of phage to the stationary culture in which the biofilm is formed, or alternatively to dialyze the phage suspension before inoculation. This is due to the possibility that additional nutrients that remain in the phage media might trigger bacteria in a biofilm state to switch to logarithmic growth. These circumstances might accentuate the killing effect of most phages. 

Further steps should be taken to promote phage therapy against biofilms. These include: developing fast and accurate techniques for the prediction and matching of phages against biofilms, and establishing a large collection of anti-biofilm specialized phages by selective enrichment. Such improvement might have extensive clinical implications. 

Though *Providencia* spp. are not defined yet as part of the ESCAPE group of challenging antibiotic resistant pathogens [25], the overuse of wide-spectrum antibiotics may result in the emergence of extreme resistance patterns and lead to more nosocomial outbreaks, which explains the growing concern among health professionals.

Currently, there are not enough available effective options to fight cases of *Providencia* biofilms infection: an extremely limited number of phages targeting *P. stuartii* and *P. rettgeri* were isolated, and no phage therapy was applied using *Providencia* phages. Thus, the results presented here open new avenues for bacteriuric long-term catheterized patients, and for the phage therapy field.

In conclusion, our results indicate that, though resistant to antibiotics, *Providencia* strains might be sensitive to phages, a promising direction to consider in the continuous struggle against the current emergence of multidrug resistant *Providencia* spp.

## 4. Materials and Methods 

### 4.1. Bacterial Strains 

In this work, 41 clinical isolates of *Providencia stuartii* and *Providencia rettgeri* were used (Table 2). A total of 10 of them were isolated locally, at the clinical microbiology lab, Hadassah-Hebrew University Medical Center, Jerusalem, Israel, and 31 strains were provided by the Center for Innovative Phage Applications and Therapeutics (IPATH) of the University of California, San Diego. Bacterial strains were grown in LB broth (Difco, Sparks, MD, USA) at 37 °C shaken on a rotary shaker at 200 rpm under aerobic conditions. Bacterial resistance or sensitivity to antibiotics was determined by the infectious disease unit of Hadassah Hospital, Jerusalem using VITEK (BioMérieux, Carponne, France).

### 4.2. Phage Isolation and Propagation 

Isolation of phages was performed using the standard double-layer agar method previously described [26] with mild modifications. Preliminary enrichment of phages was carried out by overnight growth of the bacteria in an environmental sewage sample collected from various places in Jerusalem and the surrounding. Samples were centrifuged at 7800 rpm for 10 min (centrifuge 5430R, rotor FA-45-24-11HS; Eppendorf, Hamburg, Germany). The supernatant was filtered through 0.22-μm-pore-size filters (Merck-Millipore, Cork, Ireland) and LB 5X was then added to the filtered sewage. Grown bacterial cultures of *Providencia stuartii* and *Providencia rettgeri* were mixed and inoculated with filtered sewage and LB overnight at 37 °C shaken on a rotary shaker at 200 rpm under aerobic conditions. Following inoculation, the liquid was centrifuged and re-filtered through 0.22-μm-pore-size filters. The filtered lysate was then spotted on double-layered agarose plates which were incubated overnight at 37 °C. Double-layered agarose plates were prepared by covering LB plates with 3.5 mL of pre-warmed agarose (0.25%) containing 0.1 mL of an overnight culture of bacteria (Table 2). Clear plaques were observed and purified by a minimum of four passages (of repeatedly picking single plaques and spreading them on a new bacterial lawn). Finally, single plaques were collected and transferred into bacterial liquid cultures to collect high titer lysates, which were then filtered and stored at 4 °C. The concentration of the phages was determined by counting plaque-forming units (PFU) as described [26]. Briefly, the lysate was serially diluted 10-fold into LB broth, and drops of 5 µL were spotted on overlaid tested bacterial strain lawns of agarose (0.25%), and then incubated overnight at 37 °C. The plaques were counted, and the initial concentration of PFU/mL was calculated. 

### 4.3. Phgae Genome Sequencing

The phages’ DNA was purified as described [27] using a phage DNA isolation kit (Norgen-Biotek, Thorold, Canada), and libraries were prepared with a Nextera XT DNA kit (catalog number FC-131-1096; Illumina, San Diego, CA, USA). Normalization, pooling, and tagging were performed in a common flow cell with 1 × 150 bp paired-end reads, which were used for sequencing with the Illumina NextSeq 500 platform. Sequencing was performed in the sequencing unit of The Hebrew University of Jerusalem at the Hadassah Campus. Trimming, quality control, assembly of reads, and analyses were performed using Geneious Prime and its plugins (https://www.geneious.com, accessed on 1 March 2021). Assembly was commonly performed using the SPAdes plugin of Geneious Prime. Annotation was performed using the RAST version 2 (https://rast.nmpdr.org/rast.cgi, accessed on 1 March 2021), PHAge Search Tool Enhanced Release (PHASTER) (https://phaster.ca, accessed on 1 March 2021), and BLAST servers.

### 4.4. Bacteriophages for Phylogenetic Tree

The genomes of published phages used for comparison are listed in Table 3.

### 4.5. TEM Visualization

To visualize the geometric structure of the phages, transmission electron microscopy (TEM) was used as described in OpenWetWare (http://openwetware.org/wiki/Gill:Preparingphage_specimens_for_TEM, accessed on 7 December 2017). Two ml of phage lysate with 10^9^ PFU/mL were centrifuged at 14,000 rpm (centrifuge 5430R, rotor FA-45-24-11HS; Eppendorf) for 2 h at room temperature. The pellet was resuspended in 0.2 mL of 5 mM MgSO4 and incubated overnight at 4 °C. To visualize the phage–bacteria interactions, 1 mL of lysate with 10^8^ PFU/mL was inoculated with 100 μL of overnight grown bacteria for 15 min. Then, 100 μL of the bacteria and phage mix were transferred into 100 μL of 5 mM MgSO4. On a strip of parafilm, 0.03 mL of 5 mM MgSO4 and 10 μL of the phage sample were mixed gently. Grids were prepared by placing on the drop of phage sample with the carbon side facing down for a minute, and then stain with a drop of uranyl acetate (2%) for another minute. After drying, a transmission electron microscope (Jeol, TEM 1400 plus, Peabody, MA, USA) with a charge-coupled device camera (Jeol, Gatan Orius 600) was used to capture images.

### 4.6. Assessment of Phage Lytic Activity in Planktonic Cultures

Lytic activity of the phage was assessed by inoculation of logarithmic or stationary *Providencia* cultures with phages in triplicates. The growth and lysis kinetics were recorded by measuring the optical density at a wavelength of 600 nm at 37 °C with 5 s shaking every 20 min using a 96-well plate reader (Synergy; BioTek, Winooski, VT, USA).

### 4.7. Assessment of Phage Lytic Activity in Biofilm

The 100% Latex (silicon coated) or silicone catheters (Foley Catheter) at a thickness of 16 Fr/Ch and volume of 5–15 mL were cut into ring-like pieces of ~2–3 mm width. The catheter pieces were placed in 15 mL tubes containing 3 mL LB broth with 0.1 mL of an overnight culture of bacteria. Biofilms were grown for 4 days at 37 °C shaken on a rotary shaker at 200 rpm under aerobic conditions. After transferring the catheter to a new tube and inoculation of the phage, incubation was continued overnight. Biofilms built on latex catheters were infected with PSTCR4 phage and biofilms built on silicone catheters were infected with a cocktail of PSTCR4 and PSTCR6 at the same concentration.

The catheter pieces were washed with phosphate-buffered saline (PBS), and the biomass was quantified using crystal violet staining as previously described [28]. Briefly, fixation was achieved by soaking the catheter in methanol, followed by incubation for 20 min, followed in turn by methanol aspiration and air drying. The biofilms were stained by crystal violet (1%) for 20 min at room temperature and then washed with water. The stain was extracted by adding ethanol (100%) and biomass was quantified by determining the optical density at 538 nm (OD538). In addition, the biofilm on the catheter was stained using the live stain (syto) of the Live/Dead cell viability kits (Life Technologies, Waltham, MA, USA) according to the manufacturer’s instructions. The fluorescence emission of the biofilms was detected using a confocal laser microscope (Nikon Yokogaha W1 spinning disk, Tokyo, Japan). The microscopy slices were combined to a 3D image using NIS program. For viable count, each catheter piece was transferred to 1 mL of water in 1.5 mL tubes and placed in a sonicating water bath (Bandelin Sonopuls HD 2200, Berlin, Germany) for 7 min to disrupt the biofilm, and 10 µL of 10-fold serial dilutions of each sample were plated on LB agar plates. Colonies were counted after 24 h at 37 °C and the number of colony-forming units (CFU)/mL was calculated.

### 4.8. Statistical Analysis

Prism (GraphPad, version 8.0.2 (263), San Diego, CA, USA) was used for the statistical analysis, graph formation, and design. The results were analyzed as the mean ± standard deviation in each experimental group. Statistical significance was calculated by a Student’s *t*-test two-tailed unpaired *p* values (significance level: *p* < 0.05). 

## Figures and Tables

**Figure 1 antibiotics-10-00375-f001:**
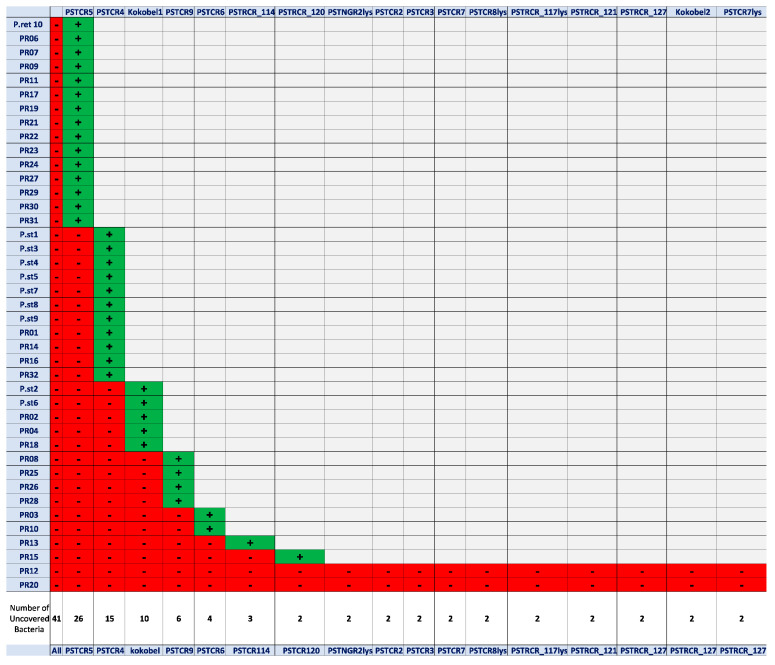
Reduction screen. The bacterial isolates killed by each noted phage (green), and those that remained for the next screen (red) after phage exposure.

**Figure 2 antibiotics-10-00375-f002:**
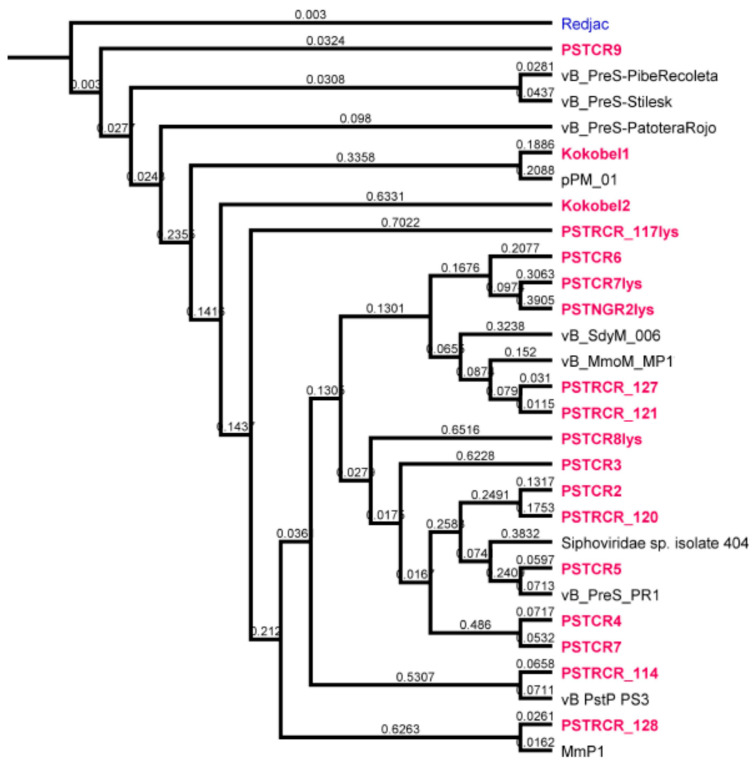
Genome analysis of *Providencia* phages. Phylogenetic tree of Providencia phages. Phages of this work are shown in red. Close-similar genomes of fully sequenced phages from the NCBI database are given as reference points shown in black.

**Figure 3 antibiotics-10-00375-f003:**
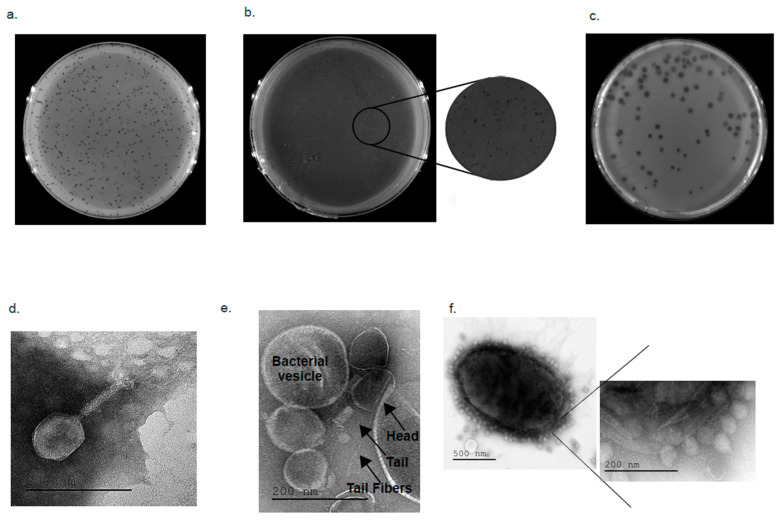
Plaque morphologies and TEM microscopy of anti-*Providencia* phages. Single plaques of PSTCR4 (**a**), PSTCR6 (**b**) on P.st 8, and PSTCR2 (**c**) on P. rett10 on bacterial lawns. Geometric structure of PSTCR4 (**d**); PSTCR6 (**e**); and PSTCR2 attached to its host P. rett10 (**f**)**.**

**Figure 4 antibiotics-10-00375-f004:**
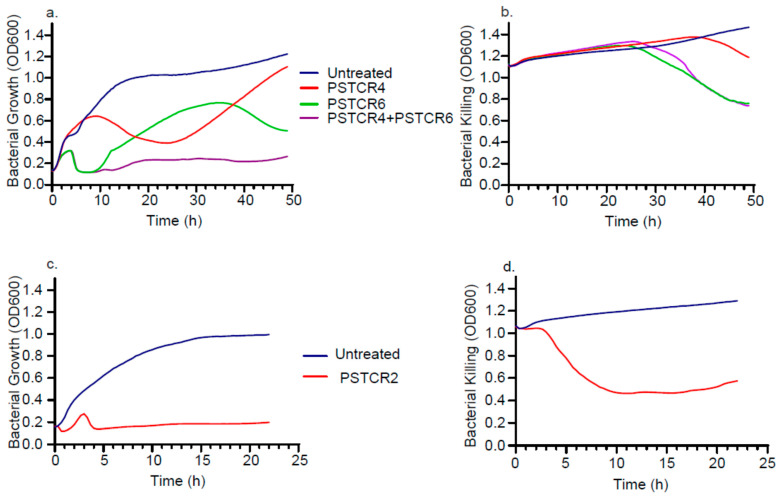
Characterization of lytic activity in liquid cultures. Kinetic growth and killing curves of P.st 8 (blue) at the (**a**) logarithmic and (**b**) stationary stage in the presence of the phages PSTCR4 (red), PSTCR6 (green) and both of them (purple); growth curves of P.rett10 (blue) at the logarithmic (**c**) and killing curves of the stationary (**d**) stage in presence of the phage PSTCR2 (red).

**Figure 5 antibiotics-10-00375-f005:**
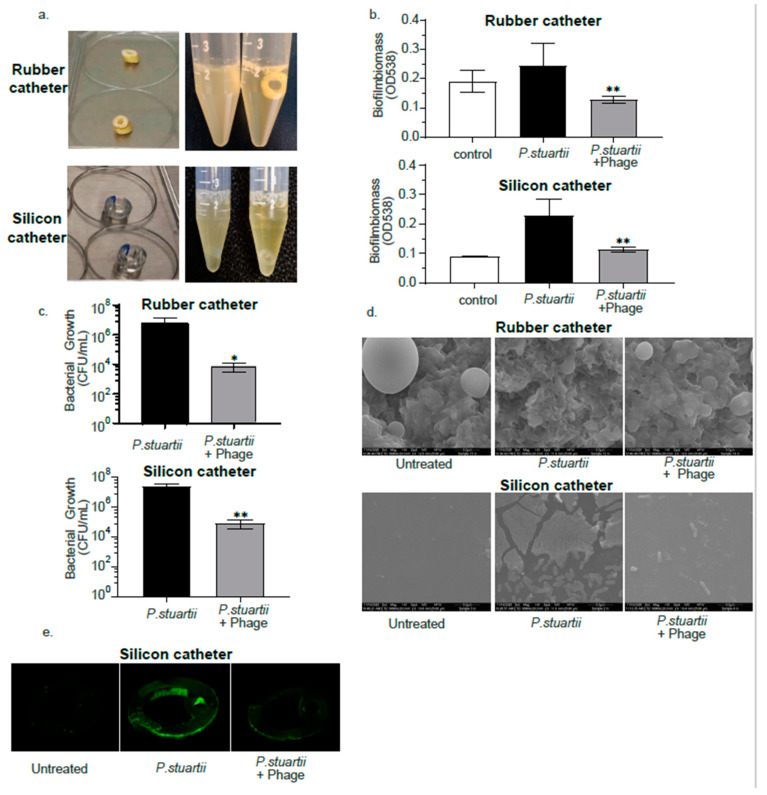
Biofilm degradation activity. (**a**) Model of *P. stuartii* biofilm formed on Latex and Silicone catheters in the absence (left tube) and presence (right tube) of phage. (**b**) Biofilm biomass quantification by crystal violet staining of *P. stuartii* formed on latex and silicone catheters; (**c**) viable colony-forming unit (CFU) counts of disrupted *P. stuartii* biofilms formed on latex and silicone catheters. (**d**) Scanning Electron Microscopy (SEM) of untreated, P.st 8 biofilm infected and phage-treated latex and silicone catheter surfaces; (**e**) confocal microscopy of untreated, P.st 8 biofilm infected and phage-treated silicone catheter biofilms. Asterisk (*, **) denotes *p*-value < 0.05 and <0.001 respectively.

**Table 1 antibiotics-10-00375-t001:** Phage collection.

Phage	Putative Life-Cycle Type ^1^	Genome Organization ^2^	Genome Size (bp)	Phylogeny	GenBank Accession
Kokobel1	lytic	circular	59,837	Siphoridae	MW145139.1
Kokobel2	lytic	circular	45,880	Siphoridae	MW145138.1
PSTNGR2lys	temperate	circular	50,958	Siphoridae	MW145137.1
PSTCR2	lytic	circular	40,200	Autographiviridae	MW057854.1
PSTCR3	temperate	circular	39,447	Myoviridae	MW057855.1
PSTCR4	lytic	circular	57,214	Siphoridae	MW057856.1
PSTCR5	lytic	circular	109,434	Siphoridae	MW057857.1
PSTCR6	lytic	circular	155,737	Myoviridae	MW057858.1
PSTCR7	lytic	circular	57,986	Siphoviridae	MW057861.1
PSTCR7lys	temperate—partial genome	linear	22,938	Siphoridae	MW057862.1
PSTCR8lys	temperate	linear	40,280	Siphoridae	MW057859.1
PSTCR9	lytic	circular	58,873	Siphoviridae	MW057860.1
PSTRCR_114	lytic	circular	42,750	Autographiviridae	MW358930.1
PSTRCR_117lys	temperate	linear	41,059	Siphoviridae	MW358929.1
PSTRCR_120	lytic	circular	39,396	Autographiviridae;	MW358928.1
PSTRCR_121	lytic	circular	165,776	Myoviridae;	MW385300.1
PSTRCR_127	lytic	circular	166,815	Myoviridae;	MW358927.1

^1^ The determination of lytic/temperate is based on the BLAST database. Cases where the full genome of the phage was found to be harbored by bacterial chromosome, served as an indication of lysogeny and were defined as temperate. Another indication for lysogeny is the similarities of the phage genes to elements involved in lysogenicity such as a lambda CI repressor, integrase, or transposon-like elements. ^2^ Linear organization refers to the constant end of the assembled genome and circular when terminal redundancy of reads was observed.

**Table 2 antibiotics-10-00375-t002:** Bacterial strains, their body site source, the provider of the isolates and antibiotic resistance.

Bacterial Strains	Source	Provided by ^1^	Antibiotic Resistance ^2^
*Providencia stuartii*			
P.st 1	blood	*HMC*	AMP, SAM, CRO, CXM, GEN, CFZ
P.st 2	blood	*HMC*	AMP, SAM, CIP, CRO, CXM, GEN, CFZ
P.st 3	blood	*HMC*	AMP, SAM, CIP, AMK, CRO, CXM, GEN, CFZ
P.st 4	blood	*HMC*	AMP, SAM, CRO, CXM, GEN, CFZ
P.st 5	blood	*HMC*	AMP, SAM, CIP, AMK, CRO, CXM, GEN, CFZ
P.st 6	blood	*HMC*	AMP, SAM, CIP, AMK, CRO, CXM, GEN, CFZ
P.st 7	blood	*HMC*	AMP, SAM, CIP, CXM, GEN, CFZ
P.st 8	blood	*HMC*	AMP, SAM, CIP, AMK, CXM, CRO, GEN, CFZ
P.st 9	blood	*HMC*	AMP, GEN
PR02	urine	*UCSD*	AMP, SAM, CXM, GEN, CFZ
PR03	urine	*UCSD*	AMP, SAM, CXM, GEN, CFZ
PR04	urine	*UCSD*	AMP, SAM, AMK, CXM, GEN, CFZ
PR12	leg lesion	*UCSD*	AMP, SAM, CIP, CFZ
PR13	leg lesion	*UCSD*	AMP, SAM, CRO, CFZ
PR14	urine	*UCSD*	AMP, SAM, CFZ
PR15	blood	*UCSD*	AMP, SAM, CIP
PR16	urine	*UCSD*	
PR25	blood	*UCSD*	AMP, CXM, GEN
PR26	blood	*UCSD*	AMP, CXM, GEN
PR32	urine catheter	*UCSD*	AMP, GEN, CFZ
*Providencia rettgeri*			
P. rett10	blood	*HMC*	AMP, GEN
PR01	breast	*UCSD*	AMP, SAM, GEN, CFZ
PR06	blood	*UCSD*	SAM, CFZ
PR07	urine	*UCSD*	AMP, SAM, CFZ
PR08	sputum	*UCSD*	AMP, SAM, CFZ
PR09	urine	*UCSD*	AMP, SAM, CRO, CFZ
PR10	heel lesion	*UCSD*	AMP, SAM, CIP, CFZ
PR11	lesion	*UCSD*	AMP, SAM, CFZ
PR17	foot culture	*UCSD*	AMP, SAM
PR18	urine	*UCSD*	AMP, SAM
PR19	urine	*UCSD*	AMP, GEN
PR20	body site, sacrum	*UCSD*	
PR21	body tissue, foot	*UCSD*	AMP, GEN
PR22	urine	*UCSD*	AMP, SAM, GEN
PR23	urine	*UCSD*	AMP, GEN
PR24	urine	*UCSD*	AMP, GEN
PR27	body site, bladder stone	*UCSD*	AMP, CXM, GEN
PR28	blood	*UCSD*	AMP, CXM, GEN
PR29	urine	*UCSD*	AMP, CXM, GEN
PR30	sputum	*UCSD*	AMP, GEN
PR31	nasopharyngeal	*UCSD*	AMP, GEN

^1^ HMC stands for Hadassah Medical center, and UCSD stands for University of California San Diego. ^2^ The antibiotics names are given in the table as abbreviations: AMP stands for ampicillin, SAM for ampicillin/sulbactam, CIP for ciprofloxacin, MEM for meropenem, ETP for ertapenem, GEN for gentamicin, CXM for ceftriaxone, CRO for cefuroxime, AMK for amikacin, and CFZ for cefazolin.

**Table 3 antibiotics-10-00375-t003:** Published genomes.

Type	Phage Name	Accession Number
Providencia phage	Redjac	NC_018832.1
Providencia phage (*P. stuartii*)	vB PstP PS3	NC_048148.1
Providencia phage (*P. stuartii*)	vB_PstP_Stuart	MK387869.1
Providencia phage (*P. rettgeri*)	vB_PreS_PR1	NC_041913.1
Providencia phage (*P. rettgeri*)	vB_PreS-PatoteraRojo	MT675126.1
Providencia phage (*P. rettgeri*)	vB_PreS-Stilesk	MT675125.1
Providencia phage (*P. rettgeri*)	vB_PreS-PibeRecoleta	MT675124.1
Morganella phage	MmP1	NC_011085.3
Morganella phage	vB_MmoM_MP1	NC_031020.1
Proteus phage	pPM_01	NC_028812.1
Shigella phage	vB_SdyM_006	MK295204.1
Siphoviridae sp. isolate 404	−	MN855780.1

## Data Availability

Sequences are available in NCBI GenBank (www.ncbi.nlm.nih.gov/genbank, accessed on 1 March 2021).

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
