# Peer review of "Targeting Biofilm of MDR Providencia stuartii by Phages Using a Catheter Model"

_antibiotics, 2021, doi:10.3390/antibiotics10040375_

Round 1

Reviewer 1 Report

Recent international reports established the emergence of multiple drug resistant (MDR) isolates of Providencia spp. The manuscript describes the isolation and characterization of 12 lytic and 5 lysogenic bacteriophages including full genome sequencing targeting multi-drug resistant clinical isolates of Providencia spp. From Israel and California. Results strongly support the potential use of these phages to eradicate biofilms of this bacterium built in a catheter model. Authors suggest that those phages could be useful as therapeutic agents.

Excellent and exhaustive work. Methods and all experiments are well described and correctly performed. Details about the phages and bacterial strains, accession numbers and so are clearly shown at Tables in the Manuscript. Figure related to Electron microscopy visualizing are also illustrative and convincing.

Minor points:

Page 3, Table 1 and Figure 1: labels the phages at Table 1 and the upper (and lower) row at Figure 1 should be reconciled. The order is different in Table/Figure, but more important, some of them do not correspond. For instance, kokobel1 and kokobel2 (Table 1) but only one kokobel at Figure 1.

2.5 degredation, replace by degradation

Author Response

Reviewer I:

Thanks for your appreciation of our work.

Our point by point answers are below in bold:

Minor points:

Page 3, Table 1 and Figure 1: labels the phages at Table 1 and the upper (and lower) row at Figure 1 should be reconciled. The order is different in Table/Figure, but more important, some of them do not correspond. For instance, kokobel1 and kokobel2 (Table 1) but only one kokobel at Figure 1.

> We revised Fig 1 and added kokbel 2 and PSTCR7lys. The reason they were not included before was that they are irrelevant for the coverage and we wanted to save space.

About the order, Table 1 is ordered by name, however, Fig 1 is ordered by the coverage, starting with the phages that provided maximal coverage downto the one that did not contributed.

2.5 degredation, replace by degradation

> Corrected

Reviewer 2 Report

This ia a straightforward paper reporting the isolation and characterization of a set of Providencia phages from sewage and a demonstration of their acton on catheter biofilms, with the goal of developing them for therapy of urinary tract infections by AMR Providencia. There are several technical problems that detract from the general quality f the presentation:

  1. table 1. I assume that DNA in particles  is listed as circular because of terminal redundancy and as linear because of  constant ends. As the authors know, This organization usually differentiates pac and cos phages. Circular vs linear is misleading, and the heading should be "organization" rather than "topology". Also "temperate" is much preferred over "lysogenic", especially since lysogeny was not demonstrated. What is meant by "full genome analysis revealed that it is part of a bacterial genome"? Phage taxonomy seems to be a bit of a mess, and most readers will not be familiar with many of the items. It would be easier if only the major 3 types were used: myoviridae, siphoviridae and podoviridae, unless the authors would like to provide definitions  or at least references, for the other types listed.
  2. Fig. 3 Plaques are not visible in B; phage is not visible in E
  3. Fig. 4. These plots of growth in microtiter plates are confusing and unsatisfactory, do not add to the presentation and are unnecessary. Why would one ever start with stationary phase cells, in which phages do not replicate? If the authors wish to show killing curves, they should do full-size aerated shake cultures, list the phage MOIs, and plot the curves with semi-log ordinates. It would be useful to determine survival with an MOI of 3, and a starting cell density of ~108 exponential cells, in which the classical expectation is   ~95% lysis. One-step growth curves would also be  informative. A series of spot-tests with dilutions of each of the phages wth a panel of typical host strains would be highy informative.  (second paragraph of section 2.4 needs a separate heading). 
  4. Fig. 5E - top panel makes no sense why is the catheter stained without bacteria bu notifier phage have disrupted the biofilm? (this part of the figure could be omitted without detracting from the paper.

Author Response

Reviewer II:

Thanks for your constructive comments

Our point by point answers are below in bold:

  • table 1. I assume that DNA in particles  is listed as circular because of terminal redundancy and as linear because of  constant ends. As the authors know, This organization usually differentiates pacand cos phages. Circular vs linear is misleading, and the heading should be "organization" rather than "topology". Also "temperate" is much preferred over "lysogenic", especially since lysogeny was not demonstrated.

> We agree with the reviewer and thank them for bringing this to our attention. We changed the titles to “Genome Organization” and in the the table to: “Temperate”. We addressed that by adding comment 2 to table 1 at Line 90.

What is meant by "full genome analysis revealed that it is part of a bacterial genome"?

>  We meant that when we find, using BLAST, the phage genomes in bacterial chromosomes, it is an indication for lysogeny. We clarify that in line 85: “The determination of lytic/ temperate is based on BLAST database. Cases where the full genome of the phage was found to be harbored by bacterial chromosome, served as an indication of lysogeny and were defined as temperate”

Phage taxonomy seems to be a bit of a mess, and most readers will not be familiar with many of the items. It would be easier if only the major 3 types were used: myoviridae, siphoviridae and podoviridae, unless the authors would like to provide definitions  or at least references, for the other types listed.

> Corrected. We left only the major types.

  • Fig. 3 Plaques are not visible in B; phage is not visible in E

> Since the plaques in B are tiny we added a magnification of a region of the plate. We also added arrows to fig 3E explaining phage structure and we hope both are clearer now.

  • Fig. 4. These plots of growth in microtiter plates are confusing and unsatisfactory, do not add to the presentation and are unnecessary. Why would one ever start with stationary phase cells, in which phages do not replicate? If the authors wish to show killing curves, they should do full-size aerated shake cultures, list the phage MOIs, and plot the curves with semi-log ordinates. It would be useful to determine survival with an MOI of 3, and a starting cell density of ~10exponential cells,in which the classical expectation is   ~95% lysis. One-step growth curves would also be  informative. A series of spot-tests with dilutions of each of the phages wth a panel of typical host strains would be highy informative.  

(second paragraph of section 2.4 needs a separate heading). 

> We thank the reviewer for that. The numbering of sections was corrected and the headline was added (Line 148 of the revised manuscript).

  • Fig. 5E - top panel makes no sense why is the catheter stained without bacteria but notifier phage have disrupted the biofilm? (this part of the figure could be omitted without detracting from the paper.

> As the reviewer suggested, we omitted the latex catheter confocal microscopy figure from Fig 5e and the text was changed accordingly (Line 202).